# References Values of Soluble α-Klotho Serum Levels Using an Enzyme-Linked Immunosorbent Assay in Healthy Adults Aged 18–85 Years

**DOI:** 10.3390/jcm11092415

**Published:** 2022-04-25

**Authors:** Andrea Espuch-Oliver, Héctor Vázquez-Lorente, Lucas Jurado-Fasoli, Tomás de Haro-Muñoz, Irene Díaz-Alberola, María del Señor López-Velez, Teresa de Haro-Romero, Manuel J. Castillo, Francisco J. Amaro-Gahete

**Affiliations:** 1Unidad de Gestión Clínica de Laboratorios Clínicos, Hospital H.U. San Cecilio, Ibs. Granada, Complejo Hospitalario de Granada, 18016 Granada, Spain; andreaespuch@gmail.com (A.E.-O.); tomas.haro.sspa@juntadeandalucia.es (T.d.H.-M.); msenor.lopez.sspa@juntadeandalucia.es (M.d.S.L.-V.); mariat.haro.sspa@juntadeandalucia.es (T.d.H.-R.); 2Department of Physiology, University of Granada, 18071 Granada, Spain; juradofasoli@ugr.es (L.J.-F.); mcgarzon@ugr.es (M.J.C.); 3PROmoting FITness and Health through Physical Activity Research Group (PROFITH), Department of Physical Education and Sports, Faculty of Sport Sciences, University of Granada, 18007 Granada, Spain; 4Servicio de Análisis Clínicos e Inmunología, Unidad de Gestión Clínica, Laboratorio Clínico, Hospital Universitario Virgen de las Nieves, 18014 Granada, Spain; idiazalberola@gmail.com; 5Instituto de Investigación Biosanitaria de Granada (ibs.Granada), 18016 Granada, Spain

**Keywords:** α-Klotho protein, aging, ELISA, references values

## Abstract

α-Klotho protein is a powerful predictor of the aging process and lifespan. Although lowered circulating soluble α-Klotho levels have been observed in aged non-healthy individuals, no specific reference values across a wide range of ages and sex using an enzyme-linked immunosorbent assay (ELISA) are available for larger cohorts of healthy individuals. The present analytical cross-sectional study was aimed to establish the reference values of soluble α-Klotho serum levels in healthy adults by age and sex groups. A total of 346 (59% women) healthy individuals aged from 18 to 85 years were recruited. Subjects were divided by sex and age as: (i) young (18–34.9 years), (ii) middle-aged (35–54.9 years), and (iii) senior (55–85 years) individuals. The soluble α-Klotho levels were measured in serum using ELISA. Senior adults were the age-group that presented the lowest soluble α-Klotho serum levels (*p* < 0.01), with age showing a negative association with soluble α-Klotho serum levels (*p* < 0.001). No differences between sexes were observed. Therefore, soluble α-Klotho levels were especially decreased—regardless of sex—in our cohort of healthy individuals because of the physiological decline derived from the aging process. We recommend routine assessments of soluble α-Klotho levels using ELISA as a simple and cheap detectable marker of aging that improves quality of life in the elderly.

## 1. Introduction

The Klotho protein is currently considered a powerful predictor of the aging process and lifespan [1]. The Klotho gene encodes Klotho protein, which is widely expressed in the kidney, parathyroid, and brain [2]. Three subfamilies of Klotho have been described (i.e., α-Klotho, β-Klotho, and γ-Klotho) [3]. Klotho—generally referred to α-Klotho—can be found in a transmembrane-bound form [4] or a soluble form derived from the proteolytic cleavage of transmembrane Klotho [5]. Soluble α-Klotho is present in blood, urine, and cerebrospinal fluid [6]—its circulating half-life is estimated to be 7.5 h—[7]. It exerts multiple physiological functions (e.g., the attenuation of cellular oxidative stress, suppression of chronic inflammation, and regulation of mineral homeostasis) [8,9] and is considered an autocrine, endocrine, and paracrine agent [10]. Aging is associated with a progressive reduction of both Klotho gene expression and, subsequently, the circulating concentrations of Klotho protein [11]. Soluble α-Klotho levels are mainly derived from a renal origin [12], with further alterations of this organ being one of the main causes of age-related decreased Klotho levels [13]. Sex has been shown to affect circulating soluble α-Klotho levels in animals [14]. However, sex did not significantly affect soluble α-Klotho in the early stages of life [15] and in unhealthy individuals [16], suggesting the influence of sex on circulating soluble α-Klotho levels to be controversial.

Although previous studies have demonstrated that the aging process is related to lowered soluble α-Klotho levels in non-healthy individuals [11,17], the lack of available and reliable investigations assessing circulating values of soluble α-Klotho in healthy individuals by age and sex is one of the main problems in establishing its reference values [18]. In this regard, the enzyme-linked immunosorbent assay (ELISA) test has been suggested to be a useful tool for the validation of soluble α-Klotho measurements not only in healthy individuals but also in patients (e.g., diabetes or chronic kidney disease (CKD)) [19]. Similarly, the ELISA assay has been proposed to be an excellent and useful method for assessing soluble α-Klotho related functions—especially those related to mineral metabolism—in healthy volunteers [17]. Reference values are needed to identify premature physiological/metabolic senescence since aging-related cellular damage may occur at an early age and may not be related to chronological age [20].

Therefore, this study aimed to establish the reference values of soluble α-Klotho serum levels using ELISA in 346 healthy adults aged 18–85 years by age and sex groups. We hypothesized that soluble α-Klotho serum levels are affected by age and sex in our cohort of healthy individuals of southern Spain.

## 2. Materials and Methods

### 2.1. Subjects and Study Design

Healthy volunteers (*n* = 346, 143 males) aged from 18 to 85 years participated in the present analytical cross-sectional study with no medical conditions and according to specific inclusion and exclusion criteria (Table 1).

The study design is the usual one for establishing the reference values of a specific biomarker. The original scheme of its description was proposed by Siest and Wilding [21], which consists of obtaining a blood sample of a representative target population (i.e., healthy Caucasian volunteers) that present a Gaussian distribution. The present study was conducted following the last revised Declaration of Helsinki and approved by the Human Research Ethics Committee of the Junta de Andalucía (0838-N-2016).

### 2.2. Blood Sample Assessment

Blood samples were requested from the biobank of the Andalusian Public Health System, whose coordination node is located in Granada (Spain). All blood samples were collected between 2017 and 2020. Since the study has no direct clinical implications and the volunteers were anonymous, no more informed consent was required in addition to those obtained from the biobank.

The blood samples were obtained from the antecubital vein after overnight fasting for 12 h and in resting conditions (at least 10 min before) in a supine position and were collected in vacutainer tubes 13 × 75 mm (Vacutainer SST, Becton Dickinson, Plymouth, UK). Serum was acquired by centrifugation (i.e., 4 min at 3000 rpm), aliquoted, and immediately stored frozen at −80 °C in the biobank. We received a total of four frozen aliquots in Eppendorf tubes (1.5 mL, Eppendorf SE, Hamburg, Germany) with a volume of 0.350 mL each (1.4 mL per volunteer).

#### Soluble α-Klotho Determination

The soluble α-Klotho levels were measured in a serum sample using a solid-phase sandwich (ELISA; kit reference number: JP27998; IBL International GmbH laboratories, Hamburg, Germany) according to the manufacturer’ protocol.

The ELISA method is based on two specific antigen-antibody reactions and semi-quantitative measurement by spectrophotometry. An ELISA plate contains 96 precoated wells with a purified mouse anti-human (IgG) soluble α-Klotho monoclonal antibody (67G3). During the assay, 100 µL of the prediluted calibrators, controls, and diluted samples from the volunteers were added to separate wells, allowing the soluble α-Klotho protein to bind to pre-immobilized anti-soluble α-Klotho antibodies in a first incubation at 23 °C for 60 min. Two consecutive washes were performed (i.e., PBS-Tween20 0.05% and pH 7.4 wash buffer) to eliminate the unjointed sample fraction to the immobilized antibody. Then, 100 µL of a second HRP-conjugated mouse anti-human (IgG) soluble α-Klotho monoclonal antibody (91F1) was added and incubated at 23 °C for 30 min, allowing the second binding of the HRP-conjugated antibody to the soluble α-Klotho protein. Two additional washes were conducted to remove the excess of HRP-conjugated anti-human soluble α-Klotho antibody, subsequently adding a chromogenic substrate (i.e., 100 µL of the enzyme TMB (3,3′,5,5′-tetramethylbenzidine)). After a further incubation at 23 °C for 30 min, 100 µL of the stop solution (1 N H_2_SO_4_) was added. Enzyme activity was measured by a spectrophotometer at 450 nm after the addition of the stop solution (i.e., 30 min).

The concentration of soluble α-Klotho was determined from a standard curve built by serial dilution. Optical densities were registered at 450 nm and converted to concentration units in pg/mL through the TRITURUS^®^ analyzer hardware and using a four-parameter logistic regression to extrapolate the concentration of the samples from the standard curve. The intra- and inter-assay coefficients of variation were calculated by measuring two different doses with purified soluble α-Klotho protein. Both coefficients of variation ranged from ~3% to ~10%. All samples were measured in one run (in the same assay batch), and blinded quality control samples were included in the assay batches to determine laboratory error in the measurements.

### 2.3. Statistical Analysis

All variables are expressed as means, standard deviations, and reference intervals (5th–95th percentile). The normality of the present data was confirmed by the Kolmogorov–Smirnov test and Q-Q plots. Subjects were categorized based on their age as: (i) young (18–34.9 years), (ii) middle-aged (35–54.9 years), and (iii) senior (55–85 years) individuals.

A one-way analysis of variance was applied to examine differences in soluble α-Klotho serum levels across the age categories. An unpaired Student’s *t*-test was used to evaluate differences between the sexes on α-Klotho serum levels in each specific age category. A single linear regression analysis was also conducted to study the association of age with the soluble α-Klotho serum levels.

The statistical analysis was conducted using the Statistical Package for the Social Sciences (SPSS v.24, Inc. Chicago, IL, USA), and plots were built using GraphPad Prism 8 (GraphPad Software, San Diego, CA, USA). Significance was set at *p* ≤ 0.05.

## 3. Results

### 3.1. References Values of Soluble α-Klotho Serum Levels

Table 2 shows the soluble α-Klotho serum levels of the 346 healthy participants (58% women) that were considered to determine references values of soluble α-Klotho by sex and across age groups ranged 18–85 years (i.e., 18–34 years (young group), 35–54.9 years (middle aged group), and 55–85 years (senior group)). We had two times more women than men in the 18–34.9 years age group. The soluble α-Klotho concentrations ranged from 60 to 3309 pg/mL.

### 3.2. Soluble α-Klotho Serum Levels Differ across Age but Not by Sex

Differences for soluble α-Klotho serum levels across the three above-mentioned age groups, for all participants, and by sex can be found in Figure 1. Soluble α-Klotho serum levels decreased significantly between young and both middle-aged (−14.6%) and senior adults (−34.4%) (*p* = 0.020 and *p* < 0.001, respectively; Figure 1A). Soluble α-Klotho serum levels in middle-aged adults were also significantly higher (30.2%) than those obtained in senior adults (*p* = 0.002; Figure 1A). Moreover, soluble α-Klotho serum levels were similar between sex in each age group (all *p* > 0.08), as shown in Figure 1B.

### 3.3. Age Is Inversely Related to Soluble α-Klotho Serum Levels

Finally, there was a significant negative association of age with soluble α-Klotho serum levels in our study cohort (β = −0.300; R^2^ = 0.151; *p* < 0.001; Figure 2).

## 4. Discussion

In the present study, we provide reference values of soluble α-Klotho levels, specifically in a considerable cohort of healthy adults in different age groups from a wide age range and in both sexes. As hypothesized, α-Klotho is importantly influenced by age, obtaining its lowest levels in the senior group, but, interestingly, no differences by sex were observed.

Previous scientific studies performed the ELISA test for the measurement of circulating soluble α-Klotho levels in healthy individuals (i.e., age ranging from 0.1–88 years) [17]. As observed in our study (conducted in a larger cohort of participants), differences by age but not by sex were detected, although lower mean levels of soluble α-Klotho (i.e., 562 ± 146 pg/mL) were observed in their recruited subjects [17]. Likewise, a population aged (0–91 years) showed declined circulating α-Klotho levels—tested with ELISA—while age decreased, although no differences by sex were observed [22]. When ELISA tests are performed in unhealthy populations living with diabetes or CKD, soluble α-Klotho concentrations tend to decrease due to potential interferents such as bilirubin or glycated hemoglobin [19]. The ELISA test has been compared with the use of immunoprecipitation–immunoblot (IP–IB) for the analysis of circulating soluble α-Klotho levels in patients who suffer from CKD. Neyra et al. suggested IP–IB as a more appropriate method to analyze soluble α-Klotho in these subjects, showing a better recovery (capture) of Klotho protein and less susceptibility to variability from sample additives when compared to the ELISA test [23]. However, considering its labor-intensive nature and high-throughput work, the ELISA test is usually preferable since it is a faster, easier, and more useful method in the routinary analytical determination of soluble α-Klotho [23]. Pedersen et al. [18] also compared two different immunoassays (i.e., time-resolved fluorescence immunoassay (TRF) vs. ELISA) aimed at determining the soluble α-Klotho levels in a total of 120 healthy individuals aged 19–66 years. TRF tended to reflect higher soluble α-Klotho levels than those provided by ELISA, although no correlation between the assays was observed [18]. Again, soluble α-Klotho differences by age but not by sex were reported when data were obtained by ELISA test, although we obtained mean soluble α-Klotho values almost two-fold higher [18]. The differences observed for soluble α-Klotho values in the previous studies may be due to undiagnosed concomitant diseases or disorders that may be related to a decrease in soluble α-Klotho levels.

In the present study, we show a negative relationship of age with soluble α-Klotho levels obtained by ELISA, with the youngest group showing the highest dispersibility of circulating soluble α-Klotho. Only two previous studies have reported age differences in soluble α-Klotho values using the ELISA test [17,18], with this association being stronger when the age ranges of the enrolled individuals are more pronounced. Soluble α-Klotho is highly expressed in the kidney and the brain [24]. The kidneys are importantly affected by age-related tissue damage (i.e., renal atrophy, glomerulosclerosis, and tubulointerstitial fibrosis), increasing the risk of CKD and thus decreasing soluble α-Klotho expression [25], which may be explained by the loss of functionality of the nephron cells involved in its expression. Furthermore, the brain is also affected by age-related changes that may suppose a downregulation of α-Klotho expression that is potentially caused by neurodegenerative diseases [26]. Therefore, it seems clear that the implementation of specific strategies to maintain an appropriate expression of Klotho in the brain and kidneys is needed to extend the life span [27]. Moreover, aging-related disturbances (e.g., increased oxidative stress, chronic inflammation, and adiposity) could modulate soluble α-Klotho levels [8] by damaging blood vessels and the histological structures of micro-vessels which are very similar in the kidneys and brain, thus reducing the expression of Klotho protein [28]. A potential alternative approach for that purpose could be the use of recombinant α-Klotho’ forms or to administer specific molecules to increase the expression of all forms of Klotho [29]. Importantly, non-pharmacological strategies have been proposed as effective ways to increase soluble α-Klotho levels, especially with advancing age (i.e., moderate or high-intensity exercise or a low-calorie, high-protein diet) [29]. Finally, the dispersibility of circulating soluble α-Klotho observed in the youngest group might be due to the lower tendency to present deficient soluble α-Klotho (e.g., young condition or practicing physical exercise) [30,31]. However, further studies are necessary to test these hypotheses.

The implementation of easily detectable biomarkers of aging is a relevant tool for monitoring the aging process and treatment, as the aging population is increasing worldwide [32]. α-Klotho protein has been described as an extremely sensitive and early marker in CKD, as its levels reflect the degree of renal insufficiency, and it can be used as an indicator of CKD progression [33]. α-Klotho (i) may be also a novel biomarker and potential treatment target for diabetic-related complications [34] and (ii) may act as a tumor suppressor and modulator of carcinomas (e.g., hepatocellular and ovarian carcinomas), representing a potential biomarker for their diagnosis [35,36]. Furthermore, a recently published study suggests α-Klotho protein as a useful biomarker in the diagnosis and prognosis of respiratory diseases (e.g., bronchopulmonary dysplasia and pulmonary hypertension) in young populations [5]. Similarly, it has been proposed as a marker of growth hormone deficiency in children with growth impairments [37].

As mentioned above, soluble α-Klotho protein can be considered as a useful biomarker in the diagnosis and prognosis of several pathologies and disorders, given its wide field of knowledge (i.e., research and clinical practice). As aged populations are increasing worldwide, simple and cheap routine analytical methods are necessary in clinical practice (i) to identify key aging markers (e.g., soluble α-Klotho) deficiencies and (ii) to improve their diminished status [33]. Altogether, this strategy can serve not only to maintain and improve the quality of life in the elderly, but also to decrease the economic cost overruns to health care systems derived from pathologies that imply low levels of S-Klotho (e.g., neurological disorders, CKD, and cancer) [38,39]. We, therefore, recommend the implementation of routine analytics of soluble α-Klotho protein in order to detect α-Klotho alterations in populations at risk of α-Klotho protein deficiency.

The present study’ findings should, however, be taken with caution as some limitations arise. First, we did not measure the intracellular form or the cell-membrane form of the α-Klotho gene which would have enriched the study. However, their assessment is very expensive and complex, making their implementation impossible in routine analysis. Second, we did not compare the α-Klotho levels obtained through the ELISA test with other analytical methods to check whether the above-mentioned reference values apply to them. Third, the results were not controlled for indices of kidney function (e.g., creatinine clearance or cystatin C). Last, α-Klotho serum levels may not sufficiently reflect tissue concentrations of the α-Klotho protein, which cannot be obtained for analysis in the absence of a clinical indication for muscle biopsy.

## 5. Conclusions

The present study provides, for the first time, soluble α-Klotho reference values in healthy adults with a wide age range (i.e., 18–85 years) by sex using the ELISA test. As predicted, age was negatively associated with soluble α-Klotho levels as a consequence of the physiological decline derived from the aging process. Interestingly, soluble α-Klotho values were similar by sex. We, therefore, recommend routine assessments of soluble α-Klotho levels using ELISA as a simple and cheap detectable marker of aging that should be checked and enhanced in cases of deficiency, especially in those areas with a high tendency to present aged populations, thus enhancing quality of life in the elderly.

## Figures and Tables

**Figure 1 jcm-11-02415-f001:**
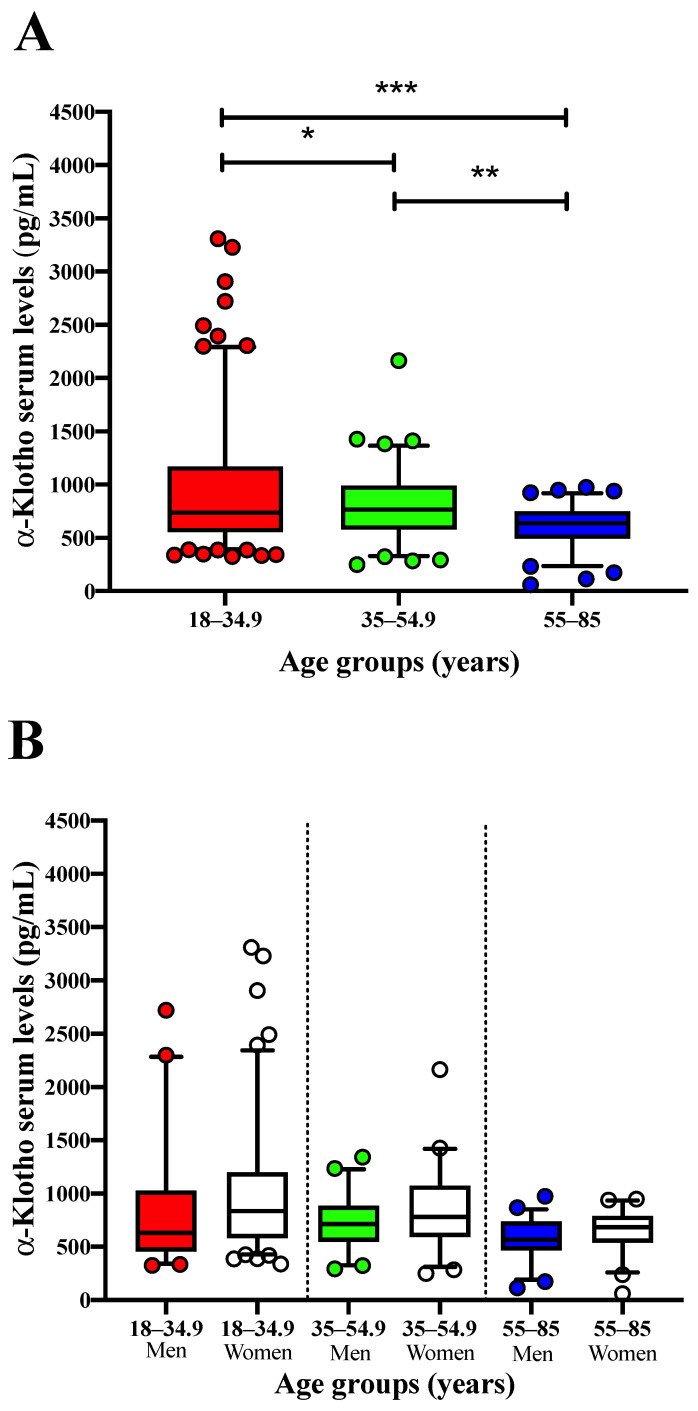
Soluble α-Klotho serum levels across age groups (i.e., 18–34 years (*n* = 167) vs. 35–54.9 years (*n* = 88) vs. 55–85 years (*n* = 91)) for all participants (**A**) and segmented by sex (**B**). (**A**): One-way analysis of variance to compare soluble α-Klotho serum levels across age groups (* *p* < 0.05, ** *p* < 0.01, and *** *p* < 0.001). (**B**): Student’s unpaired *t*-test including sex as a fixed variable (men vs. women) to determine differences in soluble α-Klotho serum levels in each age group. Values are expressed as means ± reference interval (5th–95th percentile).

**Figure 2 jcm-11-02415-f002:**
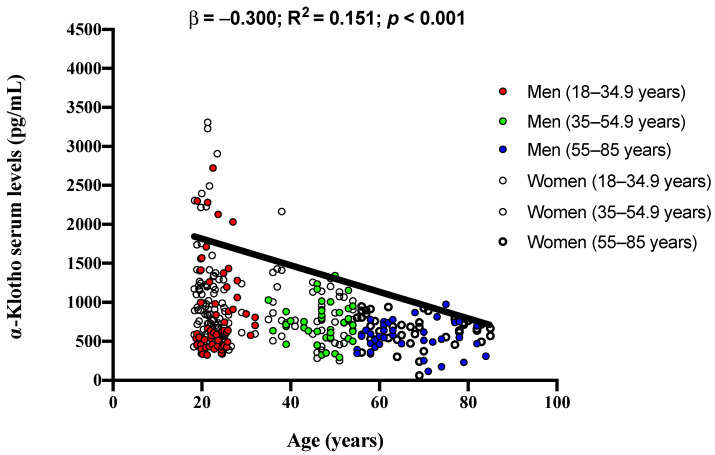
Association of age with soluble α-Klotho serum levels in adults aged 18–85 years. β (standardized regression coefficient), R^2^, and *p* values are for simple linear regression analyses.

**Table 1 jcm-11-02415-t001:** Inclusion and exclusion criteria.

Inclusion Criteria	Exclusion Criteria
- Southern Spain (Andalusian) healthy blood donors	- Cardiovascular disease
- Age ≥ 18 years	- Diabetes
- Same proportion of men and women	- Pregnancy
- Caucasian race	- Severe or poorly controlled hypertension
	- Nutritional deficits
	- Kidney disease, dialysis, or kidney transplant
	- Liver disease
	- Diseases or any condition that may interfere with bone metabolism (hyper/hypoparathyroidism, hyperthyroidism, chronic alcoholism, digestive surgery, inflammatory bowel disease, treatment with lithium, or thiazide diuretics)
	- Chronic/recurrent infectious diseases
	- Autoimmune diseases
	- Personal or first-degree family history of cancer

**Table 2 jcm-11-02415-t002:** References values of soluble α-Klotho serum levels using an enzyme-linked immunosorbent assay in healthy adults aged 18–85 years.

α-Klotho (pg/mL)				
	Mean	SD	Reference Interval (5th–95th Percentile)	
All (*n* = 346)	813.8	461.2	340.1	1672.5
Men (*n* = 143)	728.3	412.0	323.4	1541.8
Women (*n* = 203)	874.0	485.0	385.4	1750.8
18–34.9 years (*n* = 167)	932.6	575.6	392.6	2291.8
Men (*n* = 57)	850.9	566.4	341.2	2282.8
Women (*n* = 110)	975.9	578.3	427.2	2345.5
35–54.9 years (*n* = 88)	796.7	317.4	330.2	1364.6
Men (*n* = 41)	733.9	249.6	324.6	1228.3
Women (*n* = 47)	851.5	360.2	312.6	1420.0
55–85 years (*n* = 91)	612.1	198.2	235.8	918.6
Men (*n* = 45)	567.9	192.5	190.4	851.5
Women (*n* = 46)	655.5	196.2	260.4	934.4

Data are expressed as means, standard deviations (SD), and reference values (5th–95th percentile).

## Data Availability

The data that support the findings of this study are available from the corresponding author upon reasonable request.

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
