# Peer review of "References Values of Soluble α-Klotho Serum Levels Using an Enzyme-Linked Immunosorbent Assay in Healthy Adults Aged 18–85 Years"

_jcm, 2022, doi:10.3390/jcm11092415_

Round 1

Reviewer 1 Report

1- the type of study should be mention in both abstract and material and method sections.

2- How the age and  diseases could  affect  a-Klotho  synthesis and levels? probably relevant mechanisms should be more explained?

3- what is its circulation  half life ? should be pointed.

Author Response

April 21st, 2022

Journal of Clinical Medicine

Manuscript ID: jcm-1668723

Title: References values of soluble a-Klotho serum levels using an enzyme-linked immunosorbent assay in healthy adults aged 18-85 years

Dear Editor,

We are pleased to learn that our manuscript “References values of soluble a-Klotho serum levels using an enzyme-linked immunosorbent assay in healthy adults aged 18-85 years” by Espuch-Oliver et al. may be suitable for publication in the Journal of Clinical Medicine after major revision.

We deeply appreciate the accurate review process of our paper and express our gratitude to the reviewers for the extremely valuable comments and suggestions they provided. We have given them our careful consideration while revising our manuscript and we have addressed all their comments and suggestions as can be seen in our reply to the reviewers. All changes made in our manuscript are clearly apparent in the revised version. We have also made sure that it is in conformity with the Journal of Clinical Medicine’ author guidelines.

Without any doubt, the required revision has enriched and strengthened the presentation of our research in the manuscript.

We hope with this revision our manuscript is acceptable for publication in the Journal of Clinical Medicine.

We look forward to receiving your decision and would like to express our gratitude
once again for your interest in our research.

Kindest regards,

Dr. Héctor Vázquez Lorente & Francisco J. Amaro-Gahete

Department of Physiology, Faculty of Medicine, University of Granada, Granada, Spain.

E-mail: hectorvazquez@ugr.es & amarof@ugr.es

Reviewer 1:

Comment 1: the type of study should be mentioned in both abstract and material and method sections.

Authors’ reply 1: Following the Reviewer’ comment, we have addressed the type of study in both the abstract and the material and method sections.

Comment 2: How the age and diseases could affect a-Klotho synthesis and levels? probably relevant mechanisms should be more explained?

Authors’ reply 2: We really appreciate the Reviewer’ comment. Accordingly, we have explained the specific age and diseases related mechanisms involved in a-Klotho synthesis and levels in the discussion section as suggested by the reviewer.

Comment 3: what is its circulation half-life? should be pointed.

Authors’ reply 3: The circulation half-life of a-Klotho protein has been included in the introduction section (i.e., 7.5h). We thank the reviewer’s suggestion.

Reviewer 2 Report

Dear Authors!

This is an interesting topic. Nevertheless, I suggest some English editing, so it would be clearer, as well as some other considerations:

  1. In the Abstract section, several statements are doubled at various locations, therefore I suggest writing the abstract more concise and straight to the point. Please make sure, that statements are not doubled.
  2. The Introduction si too long. make sure you have only three paragraphs, explaining background of the problem, details about the problem taht lead to your research and the aim. The reader needs to know, what is the problem of ageing and the relation of Klotho to it, what does the lack of reference values for Klotho mean for everyday practice and the aim of the present study...
  3. Inclusion and exclusion criteria need to be written into the smallest details, also stating the status of smoking, obesity etc. Also, a table with included participants' characteristics would be welcome.
  4. According to the results, it seems, that in the youngest group there is the biggest dispersibility of the results. this should be commented on at least in the Discussion section.
  5. The Discussion is not clear. Please make sure, that the clinical applicability of your results is clearly stated. Also define, whether klotho is a marker or an effector, as after reading, one can conclude, that measures should be taken to rise Klotho per se? Is that true? Or would the rise of Klotho tell us, that the body is in the state of better health than it was before the intervention? Please discuss. Also, the statement, that it should be used in routine assessment of health and aging, is too strong... Please discuss why, what added value would that have and what would clinical and therapeutic consequences its determination lead to...
  6. I miss the bigger perspective of the utilisation of this parameter, as seen by the authors, not in comparison with others. The methodology of determination is important, but for a clinician, the implementation of the test and its abilities to change the clinical reasoning are important. 

Best regards!

Author Response

April 21st, 2022

Journal of Clinical Medicine

Manuscript ID: jcm-1668723

Title: References values of soluble a-Klotho serum levels using an enzyme-linked immunosorbent assay in healthy adults aged 18-85 years

Dear Editor,

We are pleased to learn that our manuscript “References values of soluble a-Klotho serum levels using an enzyme-linked immunosorbent assay in healthy adults aged 18-85 years” by Espuch-Oliver et al. may be suitable for publication in the Journal of Clinical Medicine after major revision.

We deeply appreciate the accurate review process of our paper and express our gratitude to the reviewers for the extremely valuable comments and suggestions they provided. We have given them our careful consideration while revising our manuscript and we have addressed all their comments and suggestions as can be seen in our reply to the reviewers. All changes made in our manuscript are clearly apparent in the revised version. We have also made sure that it is in conformity with the Journal of Clinical Medicine’ author guidelines.

Without any doubt, the required revision has enriched and strengthened the presentation of our research in the manuscript.

We hope with this revision our manuscript is acceptable for publication in the Journal of Clinical Medicine.

We look forward to receiving your decision and would like to express our gratitude
once again for your interest in our research.

Kindest regards,

Dr. Héctor Vázquez Lorente & Francisco J. Amaro-Gahete

Department of Physiology, Faculty of Medicine, University of Granada, Granada, Spain.

E-mail: hectorvazquez@ugr.es & amarof@ugr.es

Reviewer 2:

Comment 1: This is an interesting topic. Nevertheless, I suggest some English editing, so it would be clearer, as well as some other considerations

Authors’ reply 1: We really appreciate the Reviewer comments and suggestions. The manuscript was revised by a native English speaker before the submission process. Nevertheless, it has been reviewed again for him correcting some typos.

Comment 2: In the Abstract section, several statements are doubled at various locations, therefore I suggest writing the abstract more concise and straight to the point. Please make sure, that statements are not doubled.

Authors’ reply 2: Following the Reviewer’ comment, we have rescheduled the abstract section avoiding the presence of doubled statements. We appreciate the reviewer’s suggestion.

Comment 3: The Introduction is too long. make sure you have only three paragraphs, explaining background of the problem, details about the problem that lead to your research and the aim. The reader needs to know, what is the problem of ageing and the relation of Klotho to it, what does the lack of reference values for Klotho mean for everyday practice and the aim of the present study...

Authors’ reply 3: According to the Reviewer’ point, the Introduction has been shortened and reorganized for a better understanding. Comment appreciated.

Comment 4: Inclusion and exclusion criteria need to be written into the smallest details, also stating the status of smoking, obesity etc. Also, a table with included participants' characteristics would be welcome.

Authors’ reply 4: Dear reviewer, the blood samples were obtained from the biobank of the Andalusian Public Health System (see methods section). Thus, no specific sociodemographic data on our study’ cohort are available with respect to smoking status or body composition. However, it should be highlighted that no comorbidities were present in the study’ subjects (see exclusion criteria).

Comment 5: According to the results, it seems, that in the youngest group there is the biggest dispersibility of the results. this should be commented on at least in the Discussion section.

Authors’ reply 5: This is an interesting point raised by the Reviewer. We have included a statement in discussion section regarding this interesting result.

Comment 6: The Discussion is not clear. Please make sure, that the clinical applicability of your results is clearly stated. Also define, whether klotho is a marker or an effector, as after reading, one can conclude, that measures should be taken to rise Klotho per se? Is that true? Or would the rise of Klotho tell us, that the body is in the state of better health than it was before the intervention? Please discuss. Also, the statement, that it should be used in routine assessment of health and aging, is too strong... Please discuss why, what added value would that have and what would clinical and therapeutic consequences its determination lead to...

Authors’ reply 6: We really appreciate the Reviewer’ suggestions. Firstly, we have addressed the clinical applicability of the results in the 5th paragraph of the discussion section. Secondly, we have clarified in the same paragraph that the determination of S-Klotho circulating levels should be taken to identify a-Klotho deficiencies. Finally, we have modified the statement “should be used in routine assessment of health and aging” in both abstract and conclusion sections according to the Reviewer comment.

Comment 7: I miss the bigger perspective of the utilisation of this parameter, as seen by the authors, not in comparison with others. The methodology of determination is important, but for a clinician, the implementation of the test and its abilities to change the clinical reasoning are important.

Authors’ reply 7: We have included an additional paragraph explaining the perspective of using the routinary soluble a-Klotho levels in clinical practice (see discussion section) as suggested by the Reviewer.

Round 2

Reviewer 2 Report

The authors have sufficiently improved the manuscript, thus I recommend it for publication.